# A Systematic Review of Evidence-Based Interventions to Reduce Binge Drinking

**DOI:** 10.3390/life15111709

**Published:** 2025-11-05

**Authors:** José-Antonio Giménez-Costa, Beatriz Martín-del-Río, Consolación Gómez-Íñiguez, Adrián García-Selva, Patricia Motos-Sellés, María-Teresa Cortés-Tomás

**Affiliations:** 1Department of Basic Psychology, Faculty of Psychology, University of Valencia, 46010 Valencia, Spain; jose.a.gimenez@uv.es (J.-A.G.-C.); patricia.motos@uv.es (P.M.-S.); 2Department of Behavioural Sciences and Health, University Miguel Hernández, 03202 Elche, Spain; bmartin@umh.es (B.M.-d.-R.); adrian.garcias@umh.es (A.G.-S.); 3Department of Basic and Clinical Psychology and Psychobiology, University Jaume I, 12071 Castelló de la Plana, Spain; iniguez@uji.es

**Keywords:** binge drinking, intervention, systematic review, randomized controlled trials, alcohol consumption

## Abstract

Binge drinking (BD) is defined as a pattern of alcohol consumption that results in a blood alcohol concentration of 0.08 g/dL or higher, typically achieved after consuming approximately 70 g of pure alcohol (about five drinks for men) or 56 g (about four drinks for women) within roughly two hours. It is highly prevalent among adolescents and young adults and has significant physical, psychological, and social consequences. Despite numerous interventions to reduce BD, there is limited systematic evidence on their effectiveness. This study presents a systematic review of randomized controlled trials (RCTs) evaluating interventions to reduce BD, focusing on their impact on frequency, intensity, and associated physical, psychological, or social outcomes. The review followed PRISMA 2020, and the protocol was registered with PROSPERO (CRD42024623481). A comprehensive search was conducted in multidisciplinary and specialized databases. Included studies were RCTs targeting BD in clinical or community populations of any age. Non-empirical and observational studies, or studies not specifically focused on BD were excluded. The risk of bias was assessed using RoB 2, and a total of 21 studies (N = 14,754 participants) were included, showing high variability in design, format, and theoretical basis. Brief, individual digital interventions predominated, although face-to-face group and multimodal interventions proved more effective. Eleven studies reported significant reductions in BD frequency or prevalence; fewer addressed intensity. Interventions based on motivational and cognitive behavioral approaches, especially in structured programmes with prolonged follow-up, were the most effective. Improvements were observed in psychosocial variables (e.g., negative consequences, self-efficacy, expectations, social norms, and intention to consume), but not in physical health or consumption of other substances. Most studies had a moderate risk of bias, and few demonstrated long-term effects. More robust, comparative, and longitudinal studies are needed.

## 1. Introduction

Binge drinking (BD) is defined as a pattern of alcohol consumption that raises blood alcohol concentration (BAC) to 0.08 g/dL (0.08%) or higher, typically occurring when an individual consumes about 70 g of pure alcohol (approximately five standard drinks) for men, or 56 g (about four standard drinks) for women, within approximately two hours [1,2]. In the United States, one standard drink contains roughly 14 g of pure alcohol [1]. Importantly, this definition varies across the lifespan. During early adolescence, comparable BAC levels can be reached with lower quantities of alcohol—around 42 g for girls (approximately three drinks) and 42–56 g for boys (three to four drinks) aged 14–15 years—due to differences in body composition and alcohol metabolism [3]. These developmental distinctions are particularly relevant when examining the epidemiology of BD in younger populations. It should also be noted that the literature uses other terms to describe the same pattern of consumption, such as *heavy episodic drinking* or *episodic excessive drinking*, among others [4,5].

This risky pattern of alcohol use is highly prevalent across many countries. In the United States, approximately 30% of young adults aged 18–25 report engaging in this behavior within the past month [1]. In Europe, about one in three individuals over the age of 15 (30%) report having engaged in binge drinking during the previous 30 days [6]. Among university students, the prevalence rises sharply, exceeding 50% in several European countries [7,8,9], with consumption levels often doubling the thresholds established for this behavior [10]. These high rates are likely influenced by social and contextual factors such as permissive drinking norms, reduced parental supervision, and peer pressure [11].

In South America, BD has increasingly emerged as a significant public health concern among adolescents and young adults. In Brazil, national data show a prevalence of approximately 8.1% of BD among adolescents aged 15–17 years in the past 30 days [12] and around 17% among young adults aged 18–24 years, with higher rates observed among men and individuals of higher socioeconomic status [13]. In Asia, a multi-country study conducted among university students across nine ASEAN nations found that while 80.8% were non-bingers, 12.8% were classified as infrequent and 6.4% as frequent binge drinkers. The prevalence of BD varied substantially between countries, ranging from below 2.5% in Indonesia and Malaysia to 39.1% in Thailand and 55.0% in Laos [14]. Collectively, these findings demonstrate that BD is not confined to high-income Western contexts but also constitutes a major public health issue in middle-income regions, where cultural, social, and contextual factors may influence patterns of risky alcohol consumption.

BD is associated with multiple biopsychosocial risks. Physiologically, it increases the likelihood of developing liver damage, cardiovascular disease, and neurocognitive impairments [10]. From a psychological perspective, it is often linked to symptoms of depression and anxiety, as well as a greater propensity for risky behaviors, including unprotected sex and the use of other psychoactive substances [15,16]. In the social sphere, negative repercussions on family dynamics, academic performance, and professional development have been documented [17]. In addition, recent studies indicate that this pattern of consumption also increases the likelihood of developing an alcohol use disorder later in life [18,19,20].

Given the high prevalence and serious consequences associated with BD, the last two decades have seen a notable increase in the development and implementation of interventions aimed at modifying this pattern of consumption. These interventions have taken various forms, are based on different theoretical frameworks, and have targeted multiple populations. Notable among these are those based on cognitive–behavioral approaches, normative feedback, and the use of digital technologies—such as web platforms and interactive text messages—which in some cases have shown some effectiveness in reducing BD [21]. However, the available results are often inconsistent and, for the most part, lack integration into rigorous comparative analyses. On the other hand, the inclusion of intervention models tailored to individual factors, such as personality or motivation for change, has proven to be a promising strategy for reducing the impact of BD [22]. Despite this, studies of higher methodological quality are needed to validate the results obtained.

In this context, the present research proposes a systematic review to synthesize the available scientific evidence on interventions aimed at reducing BD, focusing exclusively on studies that employ a randomized controlled trial (RCT) design. A Randomized Controlled Trial is an experimental study in which participants are randomly assigned to either an intervention or a control group to assess the causal effects of the intervention on specific outcomes. Randomization helps minimize selection bias and enhances the internal validity of the findings [23]. This will allow for an assessment of their impact on reducing the frequency and intensity of BD, as well as on related dimensions such as physical health, psychological well-being, and social functioning. The aim is to provide a comprehensive overview of the phenomenon and highlight the importance of promoting interventions based on scientific evidence.

Based on the above, this research aims to answer the following research questions:−What types of psychological interventions have been evaluated through RCTs for the treatment of BD?−How effective are these interventions in reducing the frequency and intensity of BD episodes?−To what extent do participants’ sociodemographic characteristics, such as age and gender, influence the effectiveness of psychological interventions?−What other associated effects (physical, psychological, or social) have been reported as outcomes of psychological interventions for the treatment of BD?

## 2. Materials and Methods

The methodology applied follows the PRISMA 2020 guidelines (*Preferred Reporting Items for Systematic Reviews and Meta-Analyses*), ensuring transparency and reproducibility at each stage of the process [24]. The protocol for the systematic review was registered in PROSPERO in December 2024 (ID CRD42024623481).

### 2.1. Literature Search Strategy

A comprehensive literature search was conducted in various multidisciplinary databases (Web of Science and Scopus) and specialized databases (PsycInfo, PubMed, Cochrane, Embase, and CINAHL). The search strategy was designed to identify relevant studies and combined specific terms related to BD interventions developed with randomized controlled trials. The descriptors used were “binge drinking,” “heavy episodic drinking,” “heavy drinking,” “heavy sessional drinking,” “dangerous drinking,” “risky single-occasion drinking,” “high-risk drinking,” “risky single occasion drinking,” “high risk drinking,” “excessive episodic consumption,” “frequent binge drinking,” “concentrated drinking episode,” “episodic heavy drinking,” “therapy,” “therapeutics,” “intervention,” “rehabilitation,” and “treatment.” The complete search string underwent several rounds of testing and refinement. Searches were conducted in the title, abstract, and keyword fields. No time or language restrictions were applied.

### 2.2. Eligibility Criteria (Inclusion and Exclusion Criteria)

This review focused on the study of interventions carried out in clinical or community settings relevant to the treatment of BD, using the following inclusion criteria: (1) peer-reviewed empirical studies; (2) that used randomized controlled trials in interventions for the treatment of BD; (3) that included participants of any age with BD identified using clear criteria; (4) on any type of psychological intervention aimed at treating BD; and (5) that reported results related to reducing the frequency or intensity of BD. All articles were excluded that (1) were non-peer-reviewed studies, review articles, editorials, conference abstracts, or observational studies; (2) did not use a randomized controlled trial in their design; (3) interventions that were conducted exclusively in experimental or laboratory settings without real-world applications; (4) that did not clearly include participants who engaged in BD, but rather populations with unspecified general alcohol consumption; or (5) studies that focused solely on universal prevention or did not clearly detail the characteristics of the intervention.

### 2.3. Data Extraction and Assessment of Risk of Bias

After searching the databases, duplicates were removed before beginning the selection process. This was carried out by applying the inclusion and exclusion criteria in two stages: first, by reviewing titles and abstracts, and then by analyzing the full content of the articles. The assessment was carried out by four researchers using blinded peer review, and disagreements were resolved by consensus. The following data were extracted from each study: study identification (author(s), year of publication, and country); sociodemographic data of participants (sample size, distribution by experimental and control group, gender, age); theoretical framework underlying the intervention; methodological characteristics of the intervention (mode of delivery, mode of implementation, duration, frequency, incentives for participation); main results on the effectiveness of the intervention in reducing the frequency and intensity of BD; other associated effects (physical, psychological, or social) as results of interventions for the treatment of BD. To assess the risk of bias of each study included in the review, the Risk of Bias 2 (RoB 2) tool was applied. RoB 2 is the revised Cochrane tool for evaluating the internal validity of randomized controlled trials [25]. It assesses potential sources of bias across five domains: (1) bias arising from the randomization process, (2) bias due to deviations from intended interventions, (3) bias due to missing outcome data, (4) bias in measurement of the outcome, and (5) bias in selection of the reported result.

Each domain is evaluated through a set of signaling questions, which guide the reviewer’s judgment and lead to a domain-level rating of “low risk of bias,” “some concerns,” or “high risk of bias.” The overall risk of bias for each study is then derived from the combination of these domain ratings, providing a qualitative metric of study quality and internal validity. This structured assessment allows comparison across studies and enhances the transparency and reproducibility of the review process.

Four researchers independently assessed the risk of bias for each study, resolving disagreements by consensus. Figure 1 shows the PRISMA flow diagram, which summarizes the search and screening process for each phase and shows the records kept at each stage.

## 3. Results

### 3.1. Study Selection

An exhaustive search of several databases yielded a total of 1456 results. After removing 1042 duplicates, the sample was reduced to 414 records to be reviewed and selected. After reading the title and abstract, 289 were discarded for various reasons: 96 were not peer-reviewed empirical studies; 26 did not use a randomized controlled trial; 70 did not use specific interventions in BD; 32 did not develop interventions focused on alcohol consumption; seven did not provide details of the intervention; seven were not studies conducted in humans, and 51 did not apply any intervention. Subsequently, the full text of the 125 selected studies was reviewed to determine whether they met the eligibility requirements, and a total of 104 were discarded. The reasons for exclusion were that they were not peer-reviewed empirical studies (*n* = 30); they did not use a randomized controlled trial (*n* = 18); they did not specifically target BD (*n* = 31); the intervention developed did not focus on alcohol consumption (*n* = 12); did not provide details about the intervention (*n* = 5); did not apply any intervention (*n* = 7); and did not provide any assessment of the reduction in alcohol consumption (*n* = 1). Finally, 21 studies were included in the review, which formed the basis for the final analysis.

### 3.2. Descriptive Characteristics of the Included Studies

The studies were published between 1996 and 2024, with a notable increase from 2011 onwards, during which time almost 80% of them were conducted. Except for one study from 1996, all the others were published from 2007 onwards. With regard to the country in which the research was conducted, one-third was carried out in the United States (*n* = 8), and the rest was carried out in European countries such as Germany (*n* = 2), Spain (*n* = 2), the Netherlands (*n* = 4), the United Kingdom (*n* = 4), and Sweden (*n* = 1).

Regarding the samples used, the 21 studies included a total of 14,754 participants (*SD* = 985.49), with considerable variability. A total of seven studies (33.33%) used samples of fewer than 200 people, while another seven studies (33.33%) used samples of between 200 and 500 people, three (14.29%) were carried out with samples of between 500 and 1000 people, and finally, four studies (19.05%) used samples larger than 1000 people, with the study by Hanewinkel et al. [26] having the largest sample size (*n* = 4163).

Given that all the studies included correspond to randomized controlled trials, the distribution of the sample between the experimental and control groups was analyzed. A total of 13 studies (61.90%) used a balanced distribution between both groups, with a difference of less than 10% in the sample sizes of both conditions. Three studies (14.29%) showed differences between 10% and 20% between the experimental group and the control group, and in two other studies (9.52%), this difference was between 20% and 25%. Only one study (4.76%) had a sample size difference between the experimental group and the control group of 38.30%. Lastly, two studies (9.52%) that used more than one experimental condition did not report the number of subjects assigned to each condition.

Finally, with regard to the characteristics of the sample, the lowest average age of all the studies analyzed was 14 years, while the highest was 38.5 years, with most studies focusing on adolescents and young people. Thus, six studies (28.57%) used samples consisting of adolescents (secondary school and vocational training students), with average ages between 14 and 17.3 years (*SD* = 0.73–1.30). Meanwhile, 13 studies (61.90%) used samples of young adults with average ages between 18.70 and 26.38 years (*SD* = 0.80–9.66). Of this latter group, 12 studies involved university students, while one study involved 20-year-old men from the military recruitment census. The remaining two studies (9.52%) had samples with average ages of 34.6 years and 38.5 years, respectively, without mentioning data on the standard deviation.

Finally, in terms of gender distribution, three studies focused on a single gender: two (9.52%) on men and one study (4.76%) on women. Six studies (28.57%) included low percentages of men, ranging from 17% to 35%. However, most of the studies, 11 in total (52.38%), had an average proportion of men, with values ranging from 41% to 60%. Finally, one study (4.76%) showed a high male representation, with a percentage of 72%.

### 3.3. Risk of Bias

Table 1 shows the results of the risk of bias assessment using the RoB 2 methodology [25]. Of the 21 studies reviewed, three (14.29%) had a high overall risk of bias, while 17 (80.95%) had some concerns and only one study (4.76%) was considered to have a low risk of bias.

In the analysis of randomization bias (D1), 11 studies (52.38%) were classified as low risk, six (28.57%) raised some concerns, two (9.52%) showed high risk, and another two (9.52%) did not provide sufficient information on the procedure. Many studies with some concerns indicated that allocation was random, without detailing the method used or possible allocation concealment. Only a few studies clearly described the procedure used.

Regarding the risk of bias due to deviations from the planned intervention (D2), 15 studies (71.43%) raised some concerns, five (23.81%) were classified as low risk, and one (4.76%) as high risk. The main problems found included the lack of blinding of participants and researchers, as well as non-homogeneous dropout or adherence rates between experimental groups.

Regarding the risk of bias due to missing outcome data (D3), eight studies (38.10%) presented low risk, another eight (38.10%) showed some concerns, four (19.05%) were classified as high risk, and one (4.76%) did not provide information on this issue. The main problem detected was the high dropout rate, with no imputation of missing data or evidence of sensitivity analysis. Some studies mention the use of the intention-to-treat principle; however, not all clearly specified the handling of missing data, especially if dropouts were uneven between groups.

In terms of risk of bias in outcome measurement (D4), 16 studies (76.19%) raised some concerns, two (9.52%) showed low risk, and three (14.29%) were classified as high risk. Most studies used self-reports as the sole source of information on alcohol consumption. Some included structured questionnaires or complementary scales, and a few incorporated objective methods. In high-risk studies, the instruments used were not validated or their reliability was not reported.

Finally, regarding the risk of bias due to selection of the reported outcome (D5), 12 studies (57.14%) showed a low risk of bias, while nine (42.86%) raised some concerns. The problems detected correspond to the absence of pre-specified protocols or the reporting of only the most relevant results, without clarifying whether other analyses were explored or unfavorable effects were analyzed.

However, based on the distribution of findings across the 21 studies included, there is no clear indication of publication bias. Specifically, nine studies reported no significant effects of the intervention on binge drinking, nine studies reported positive effects, and three studies reported mixed outcomes (e.g., significant effects on frequency but not on intensity of binge drinking). This balanced pattern of published results suggests that studies with non-significant or negative findings were also published and included, which reduces the likelihood that publication bias has substantially influenced the evidence synthesized in this review

### 3.4. Characteristics of Interventions

Table A1 in Appendix B shows the summarized information from the 21 studies analyzed. The following sections describe the main characteristics of interventions aimed at reducing or eliminating BD: modality/format, individual or group, type of intervention, total number of sessions and duration of each session, total intervention and follow-up, use of structured programs, and rewards.

#### 3.4.1. Format for Applying Interventions and Type of Interventions

The interventions reviewed were delivered digitally (14 studies) or in person (7 studies) [22,26,27,28,30,31,36].

Most digital interventions (*n* = 10) were implemented through web platforms that provided participants with specific instructions to follow (questionnaire, task) [11,21,32,33,35,37,38,39,40,41]. In two studies [21,42], the use of the web was combined with the sending of text messages via mobile phone. Crombie et al. [15] and Witkiewitz et al. [34] only used text messages to mobile phones, and Hedman and Akagi [29] used email. All participants, regardless of the modality, received personalized feedback.

In terms of the modality of the interventions, 16 studies were individual [11,15,21,29,30,31,32,33,34,35,37,38,39,40,41,42]; and in the other four it was group-based [22,26,27,36]. One study used both modalities [28].

#### 3.4.2. Total Number of Sessions and Duration

The interventions carried out show great variability in terms of the number of sessions and their duration. Only 15 studies specified the number of sessions. Seven conducted a single session [11,27,30,31,33,35,42]. Two conducted two sessions [22,34]. In the study by Wood et al. [28], between one and three sessions were conducted, depending on the type of intervention. Five studies conducted between three and eight sessions [36,37,39,40,41].

With regard to duration, only seven studies reported the duration of each session in minutes: 15.8 ± 5.5 min [30], 20 min [33,35], 30 min [42], 45–60 min [28], approximately one hour [40], and 90 min [22]. Carey et al. [11] reported a duration of one and a half hours, which includes the time spent on the baseline, the intervention, and a post-intervention questionnaire before follow-up. Only six studies described the total duration of the intervention in weeks, ranging from two to twelve weeks [15,21,26,29,34,41]. Five studies did not report the duration of the intervention [27,31,36,37,39].

Finally, two studies did not specify the number of sessions or the duration of the intervention. In particular, the study by Gilmore and Bountress [38] referred to previous studies describing a similar methodology [43,44], and the study by Voogt et al. [32] cited two other studies for consultation [33,45].

#### 3.4.3. Post-Intervention Follow-Up

In general, studies describe follow-up by indicating the number of sessions conducted and the time elapsed since the end of the intervention. The most common pattern is to conduct a single follow-up session, although the timing of this session varies depending on the study. Fourteen studies conducted a single follow-up session: two at two weeks [27,31]; four at one month [21,34,41,42]; one at six weeks [29]; one at three months [38]; three at four months [37,39,40]; one at six months [30]; one between 4 and 6 months [26]; and one at 12 months [36].

Three studies conducted two follow-up sessions, although with different time frames: at one and six months [32,33]; three and twelve months [15]; and six and twelve months [22]. One study conducted three follow-up sessions, at one, three, and six months [28]. One study conducted five follow-up sessions, at one, three, six, nine, and twelve months [11]. Finally, the study by Voogt et al. [35] conducted follow-up once a week for a period of six months, following this pattern: month one (four 10 min post-test sessions), month three (eight 10 min post-test sessions), and month six (Thirteen 10 min post-test sessions).

In general, the studies do not specify the duration of the follow-up session or sessions, with the exception of the study by Voogt et al. [35], mentioned above, and the study by Voss et al. [42], with a duration of 30 min.

#### 3.4.4. Application of Previously Designed Programs

In eleven studies, the intervention targeting BD was carried out following previously designed and structured programs. The Alcohol Alert program [37] was applied in three studies [37,39,40]. The BASICS program (The Brief Alcohol Screening and Intervention for College Students) [43] was used in two studies [34,38]. And three used the What Do You Drink (WDYD) program designed by the same authors who applied it [32,33,35].

Other programs, each applied to a single study, were the Klar bleiben (“Keep a Clear Head”) school prevention program [26]; and the eCHECKUP TO GO-Alcohol program (San Diego State Research Foundation) [46] in Chavez and Palfai [21]. Finally, one study [36] applied the Families: Preparing the Next Generation (FPNG) program.

#### 3.4.5. Incentives Offered to Participants

Twelve studies [11,21,26,28,33,34,35,36,37,38,41,42] reported offering some form of reward (e.g., money, gift cards, raffle) to participants, either at each stage or at the end of the entire process, including the follow-up stage.

#### 3.4.6. Theoretical Basis of the Interventions

The interventions analyzed are based on various theoretical frameworks, with a predominance of approaches focused on the motivational and cognitive–behavioral factors involved in behavioral change.

Firstly, studies based on motivational models stand out, particularly those that use Motivational Interviewing [47]. Five studies used this approach as their main basis. For example, the study by Conrod et al. [22] combined it with principles of cognitive–behavioral and personality theory. For their part, the studies by Voogt et al. [32,33,35] integrated elements of the I-Change model within the same motivational framework. White & Pohl [41], in addition to using motivational interviewing, incorporated other theoretical models such as the Theory of Planned Behavior [48], the Transtheoretical Model of Change [49], and the cognitive–behavioral approach [50]. Additionally, two studies [28,30] applied Brief Motivational Interviewing [51,52], a more concise adaptation of Motivational Interviewing. Finally, the I-Change model was identified as preferred in three studies [37,39,40].

Secondly, interventions based on behavior change models and cognitive–behavioral approaches were also identified. Gilmore and Bountress [38] and Witkiewitz et al. [34] developed programs that integrate motivational, cognitive–behavioral, social influence, transtheoretical, and relapse prevention components within the framework of the BASICS program. Similarly, the study by Crombie et al. [15] was based on the Health Action Process Approach model (HAPA) [53], which focuses on the stages of intention, planning, action, and maintenance of healthy behavior.

Research that integrated and combined different theoretical models was also identified. The study by Murgraff et al. [27] was based on a combination of theories, such as Social Cognition Theory, the Implementation Intentions Model [54], the Self-Control and Planning Model [53], as well as other constructs such as the conversion of intentions into concrete plans, the importance of situational control, and the formulation of action plans.

On the other hand, some studies are based on applied psychological theories. Arden and Armitage [31] used the Transtheoretical Model of Change [49]; Carey et al. [11] used Self-Affirmation Theory [55]; and Hedman and Akagi [29] used the Elaboration Likelihood Model [56], without incorporating motivational interviewing or normative information. In turn, Voss et al. [42] based their intervention on the theoretical approach of Behavioral Economics and the role of the value of delayed reinforcement [57]; while Williams et al. [36] based their work on Ecological Development Theory [58,59,60], which emphasizes strengthening family functioning as a means of preventing substance use in adolescents by empowering parents.

Finally, two studies [21,26] mentioned in general terms the consideration of motivational, cognitive, and behavioral aspects involved in consumption behavior, without explicitly linking to any theoretical model.

### 3.5. Effectiveness of Interventions in BD

#### 3.5.1. On the Prevalence of BD

Of the nine studies that evaluated the presence or prevalence of consumption as an outcome variable of the intervention, five showed significant differences [22,27,28,36,40].

This outcome measure was performed at different time periods. Thus, Murgraff et al. [27] obtained this significant difference compared to the control group two weeks after the intervention; Martínez-Montilla et al. [40] at four months; Conrod et al. [22] at six months; and Williams [36] at one year. Only the study by Wood et al. [28] conducted a follow-up with different time measures. The results showed that in two of the separate interventions (the interaction of both showed no effect), the prevalence of BD consumers was significantly reduced at the end of the intervention. However, this effect declined in the follow-ups carried out at one month, three months, and six months (in fact, one of the interventions—AEC—lost its difference with the control group in the last follow-up).

In one of the studies [37], no significant differences were found between the intervention and control groups in the total sample, but when analyzed by age group, those under 15 and 16 years of age showed significant decreases in their participation in BD after one and two sessions. In addition, those under 15 years of age maintained this significant effect four months after the intervention.

In the three remaining studies, no significant impact on BD reduction was observed either in the short term [15] or in subsequent follow-ups conducted at one month and six months [32,33].

#### 3.5.2. On the Frequency of BD Episodes

The most commonly used outcome measure to analyze the effectiveness of interventions on BD was the frequency of this pattern of consumption. It was used in 16 of the 21 studies analyzed. In seven, the results showed statistically significant differences [11,21,27,29,30,31,35].

Also, with regard to this variable, the studies used different time frames for measuring the effectiveness of the results. These ranged from two weeks after the end of the intervention in Arden and Armitage [31] and Murgraff et al. [27], to six months in Carey et al. [11] and Daeppen et al. [30], to four weeks in Chavez and Palfai [21] and six weeks in Hedman and Akagi [29]. Only one of the studies [35] checked the effect of the treatment at three points in time, at one month, three months, and six months, confirming in all cases a significantly lower frequency of BD in the intervention group compared to the control group.

In the remaining studies (*n* = 9), although some did obtain reductions in BD frequency in the intervention group [15,26,39], this reduction was not statistically significant compared to the control group. In two studies [41,42], the reduction in frequency was similar in both groups (intervention and control). And in the rest of the studies reviewed [32,33,34,38], the results showed no difference in the frequency of BD in favor of the intervention group.

#### 3.5.3. About the Amount of Alcohol Consumed in Each BD Episode

Another relevant outcome variable is the amount of alcohol consumed during a BD episode. Two studies analyzed the variation in the amount of alcohol consumed after the intervention [26,29]. The results showed that neither of these two interventions was effective in reducing the amount of alcohol consumed. Hedman and Akagi [29] noted that, although the average number of typical drinks consumed per episode decreased in the experimental group and increased in the control group, the final difference between the two was not significant. For their part, although Hanewinkel et al. [26] found no significant differences for the sample as a whole, the subgroup of adolescents who were already heavy drinkers significantly reduced their alcohol intake compared to their counterparts in the control group.

#### 3.5.4. Other Psychosocial and Consumption-Related Variables

This section covers psychosocial factors as well as consumption-related outcomes, including negative consequences of alcohol use, to provide a comprehensive view of the studied variables.

Finally, more than half of the studies (*n* = 11) explored whether interventions not specifically targeting BD also produced changes in other related psychosocial variables. The most frequently assessed psychosocial variable (*n* = 6) was the reduction in negative consequences resulting from consumption. Most interventions showed positive effects, with significantly lower scores in the intervention groups [11,26,28,42]. However, in two studies [21,34], no significant differences were observed between the groups.

The second most evaluated variable was the change in expectations and motives for consumption, where Gilmore and Bountress [38] and White and Pohl [41] obtained significant reductions in favor of the intervention group, while the studies by Hanewinkel et al. [26] and Wood et al. [28] reported no differences between the groups.

In the three interventions that evaluated perceptions of others’ consumption and perceived social norms [21,26,28], significant decreases were shown in favor of the experimental group. Meanwhile, in the three studies that analyzed the level of self-efficacy to refuse or control alcohol consumption, two of them [21,28] reported significant differences, while Hanewinkel et al. [26] found no difference due to the intervention.

For their part, neither of the two studies that analyzed the decrease in the consumption of other substances as an effect of the intervention in BD obtained significant results [26,30]. Regarding the decrease in the intention to consume alcohol, the study by Daeppen et al. [30] reported differences in favor of the intervention group, while in the work by Chavez and Palfai [21], no effect was observed.

Similar results are seen in studies that evaluated the behavioral protection strategies used after the intervention: they were only significant in one of the studies [42], while in another they were not [21].

The remaining variables were only analyzed in individual studies. Significant differences were observed in the increase in study hours [42] and in positive self-assessment [11]. However, no significant results were observed in other variables such as the reduction in risky behaviors (driving under the influence of alcohol, unprotected sex, etc.), attitude toward personal consumption [29], preference for delayed rewards [42], decreased anxiety levels [28], quality of life [39], and alcohol expenditure [42].

## 4. Discussion

This systematic review aims to analyze psychological intervention strategies targeting BD consumers, focusing on those evaluated under rigorous methodological standards (randomized controlled trials), in order to provide a comprehensive overview and synthesize the available evidence on their efficacy and clinical relevance.

Given the substantial heterogeneity across studies in terms of intervention type, outcome measures, and participant characteristics, a meta-analysis was deemed inappropriate. Instead, a qualitative synthesis was conducted to accurately represent the available evidence without introducing misleading or non-comparable summary estimates.

Overview of Evidence and Intervention Effectiveness

Although the number of interventions aimed at addressing BD has increased in recent decades [21], since 2007 the number of studies that meet acceptable methodological rigor has remained relatively constant. This situation highlights the need to improve the methodological quality of research in this field and underscores the advisability of interpreting the results obtained in studies with less methodological rigor with caution.

Among the interventions reviewed, brief, self-administered digital interventions stand out, possibly due to their accessibility and low cost. However, they have high dropout rates, which compromises both the internal validity of the results and their potential for generalization. In contrast, face-to-face interventions, especially group interventions with multiple sessions, show more consistent effects in terms of retention, especially when based on motivational or integrative models [22,36]. These findings highlight the need to question the extent to which the incorporation of technology, without the accompaniment of other structural or relational components, guarantees a significant impact on BD patterns and their psychosocial effects.

Another important aspect is the notable heterogeneity of the interventions, both in their theoretical basis and in their formal aspects (number of sessions, duration, and follow-up). Although motivational approaches predominate, applied directly through Motivational Interviewing or incorporated into programs such as I-Change, BASICS, or Alcohol Alert, their effectiveness is not uniform. For example, neither of the two studies that implemented the BASICS program—which integrates motivational, cognitive–behavioral, social influence, and relapse prevention components—showed efficacy in reducing the frequency of BD episodes [34,38] or in reducing the negative consequences associated with consumption [34], possibly due to the small number of sessions they included. It is reasonable to assume and suggest that modifying a drinking pattern such as BD requires longer and more sustained interventions over time, which allow for the consolidation and maintenance of the learning and strategies worked on during the intervention. Thus, we have found that interventions based on the Alcohol Alert program showed more favorable results, especially when implemented in six individual sessions of approximately one hour [37,40], while partial applications of the I-Change model, such as in What Do You Drink, do not produce significant changes [32,33].

Along the same lines, face-to-face group interventions that combine motivational approaches (such as the transtheoretical model, Motivational Interviewing, or its brief version) with cognitive–behavioral strategies (e.g., self-control, planning, conversion of intentions) and even work on parental discipline have proven to be the most effective. All of them report improvements in both the prevalence and frequency of BD episodes, regardless of whether they implement a comprehensive program or a set of specific strategies. These interventions share a focus on education about the risks of consumption and the promotion of individual reflection. The psychosocial variables addressed include: correcting perceived social norms (e.g., the belief that everyone drinks heavily) through the use of real data; modifying expectations of consumption, demystifying perceived positive effects (such as fun or disinhibition) and highlighting negative effects; strengthening self-efficacy to refuse consumption; developing social skills and autonomous decision-making; as well as promoting the capacity for reflection and critical thinking, and confronting values and behaviors to encourage change. In addition, evidence-based information on health risks is provided. Some programs, such as FPNG or Alcohol Alert, actively involve parents or educators as reinforcing agents in family and school settings to strengthen the key messages of the intervention.

Despite these advances, methodological limitations persist. Among these are the scarcity of studies comparing different theoretical frameworks for intervention, as well as the lack of replications that maintain the same theoretical framework and a uniform methodological structure. This situation makes it difficult to identify individual and social variables relevant to explaining and modifying BD. In addition, most of the studies analyzed conduct short-term follow-ups, generally limited to one or two sessions and not exceeding six months after the intervention, which prevents adequate assessment of the sustainability of changes in an intermittent consumption pattern influenced by contextual risk factors [31,35].

Another limitation concerns the variable intensity of BD episodes, which has been used to assess the effectiveness of interventions. This has been evaluated in very few studies and without significant results [26,29]. Although the specialized literature repeatedly recognizes it as a key variable for analyzing the consequences of BD [19], in practice, the frequency of episodes is often prioritized because it offers greater psychometric stability and is less susceptible to memory errors or contextual variations, such as the social environment or the type of event [39]. The memory of the exact number of drinks consumed in an episode may be affected by cognitive impairment associated with excessive consumption or by the participant’s subjective perception [26]. From a public health perspective, reducing the frequency of BD episodes decreases cumulative risk exposure and the likelihood of negative alcohol-related consequences [61]. Therefore, studies that include both intensity and frequency provide greater objectivity and accuracy in evaluating intervention outcomes.

Another aspect to consider is that some studies reported improvements in both groups, intervention and control. This suggests the possible influence of external factors, such as monitoring consumption or the very desire for change generated by participation in the study. Such effects may raise doubts as to whether the observed effects are really due to the interventions themselves or to spontaneous processes of behavioral self-regulation. Overcoming this limitation requires more sophisticated analytical frameworks capable of considering differential effects based on risk profiles, consumption trajectories, and ecological contexts.

Likewise, individual variables such as age and gender can also modulate the effectiveness of interventions, although they are rarely analyzed as moderating variables, which limits understanding of the groups for which interventions are effective. For example, college students are the most studied group due to the high prevalence of this type of risky consumption among them. This overrepresentation makes it difficult to generalize the findings to other vulnerable groups, such as minors, young adults not in school, or people in clinical settings. Jander et al. [37] observed that effectiveness may vary according to age, showing greater benefits in younger participants, underscoring the importance of early prevention given that there is greater scope for intervention on consumption habits that are not yet established [62]. Considering the gender perspective is equally relevant, given that there are marked differences between boys and girls in terms of prevalence, motives, expectations, and consequences associated with BD [63].

Additionally, a methodological aspect such as the use of rewards or incentives in some studies should also be mentioned. Although the use of rewards may encourage initial participation and improve follow-up rates, it could also introduce a motivational bias, since, for example, these rates may not reflect a real commitment to behavioral change, and therefore limit the generalization of the results to other contexts without such rewards.

In short, although the studies reviewed have contributed to outlining a general picture of the effectiveness of interventions, further progress is still needed in understanding how individual and contextual variables interact with the components of interventions to generate sustained change.

Finally, a contribution of the conducted review lies in the identification of the psychological and social mechanisms through which interventions exert their effects. Changes in motivation to change, alcohol-related expectancies, and normative beliefs constitute proximal mechanisms that drive behavior modification. These cognitive and motivational processes align with a broad body of evidence indicating that proximal factors interact with more stable personality traits—such as impulsivity and sensation seeking—to shape patterns of alcohol consumption and associated problems [64,65,66]. Moreover, research on brief alcohol interventions has shown that changes in perceived norms and expectancies often mediate reductions in risky drinking, offering a plausible explanation for their effectiveness [61,67]. Recent analyses of online interventions further reinforce this interpretation by highlighting the role of motivational and behavioral mediators in explaining intervention outcomes [68].

A deeper understanding of how these mechanisms operate together—linking cognitive change with dispositional tendencies—may guide the improvement of future prevention strategies by simultaneously targeting personality-related risk factors and modifiable cognitive processes.

Cultural and Contextual Determinants of Intervention Outcomes

Beyond individual and psychological mechanisms, the effectiveness of these interventions should also be interpreted in their broader cultural context. As highlighted in the introduction, alcohol consumption is deeply embedded in social norms and daily practices, particularly within Western societies, which can make prevention and behavior change especially challenging [69]. Comparative studies show that drinking patterns across Europe form relatively stable cultural clusters, emphasizing the need for culturally responsive approaches [70]. Because most available evidence comes from WEIRD populations—Western, Educated, Industrialized, Rich, and Democratic—generalizations to other cultural settings should be made with caution [71].

Total abstinence is often neither a realistic nor a culturally acceptable goal. For this reason, interventions should define specific and attainable targets, such as reducing the monthly frequency of binge drinking episodes, lowering peak episode intensity (for instance, typical number of drinks or estimated blood alcohol concentration), minimizing alcohol-related negative consequences, and promoting the use of protective behavioral strategies. Evidence from previous studies indicates that abstinence-based models are not always achievable or well accepted, whereas approaches focused on reducing consumption or related harms can produce substantial benefits [72].

The available findings also show a scarcity of significant results in physical health variables or in the reduction in other substance use. This suggests that the interventions analyzed tend to focus on a single behavior, without comprehensively considering associated lifestyles. This limitation is particularly important in contexts where BD coexists with other risk factors, such as anxiety, cannabis use, or unprotected sexual practices [73].

Consequently, the results linked to BD interventions should not be interpreted as secondary or marginal effects. On the contrary, they present complementary indicators that can intensify or attenuate the main behavioral impact, providing valuable information for the design of more comprehensive interventions, adjusted to multiple dimensions of the problem and, therefore, potentially more effective.

### 4.1. Strengths and Limitations

This review has methodological strengths that support the validity of its conclusions. First, its design and execution complied with the PRISMA 2020 guidelines (Appendix A) [24], with a protocol previously registered in PROSPERO, ensuring transparency, traceability, and reproducibility throughout the process. Likewise, a comprehensive search strategy was applied, without language or date restrictions, which included general and specialized databases, increasing the sensitivity and specificity of the results obtained. The systematic use of clearly defined inclusion and exclusion criteria and the assessment of risk of bias using Cochrane’s RoB 2 provided a nuanced view of the quality of the included studies.

However, this review also has relevant limitations. First, the heterogeneity of the included studies—in terms of design, population characteristics, intervention format, and outcome measures—prevented a quantitative meta-analysis from being performed, limiting the scope of comparisons between studies. Despite efforts to organize the findings systematically, variability in the instruments used and timing of assessments prevented the effects from being grouped homogeneously.

Much of the evidence was based on self-reported measures, without cross-validation through objective sources or external informants, introducing an inherent risk of measurement bias [32]. This limitation is particularly relevant in the study of BD, given that social desirability and the tendency to underestimate consumption can significantly distort self-reported data. Despite this, some previous research suggests that these measures are somewhat reliable [74].

The methodological quality of the randomized controlled trials was uneven, with frequent “some concerns” regarding allocation concealment, handling of missing data, and selection of results.

Finally, the review focused exclusively on studies published in peer-reviewed journals, which, while ensuring a minimum threshold of quality, may have left out relevant interventions found in gray literature, doctoral theses, or unpublished pilot projects. This possible publication bias could have influenced the estimation of the actual effect of the interventions.

### 4.2. Future Recommendations

To move forward, it is recommended to promote studies with greater methodological robustness, including rigorous randomization procedures, allocation concealment, and intention-to-treat analysis, as well as rigorous strategies to manage bias derived from missing data. Likewise, transparency in the pre-specification of hypotheses and results should become standard practice through the systematic use of prior registries (e.g., PROSPERO, Open Science Framework).

Secondly, it is necessary to incorporate moderation and subgroup analyses that consider individual variables—such as age and gender—as well as the socio-educational context. This would allow for the design of interventions that are more tailored and sensitive to the heterogeneity of the phenomenon, increasing both their effectiveness and their applicability in the specific circumstances of each group. It is also recommended to conduct comparative studies between different intervention formats, particularly those that contrast brief digital approaches with more intensive, group, or face-to-face interventions. Such studies will help to determine not only which interventions are effective, but also for whom, in what context, and at what level of intensity.

In addition, it is suggested that the focus be broadened beyond alcohol consumption to include indicators of psychosocial change, well-being, and quality of life, and to consider cultural and social norms that influence consumption. This would allow for a more accurate understanding of the psychological mechanisms underlying behavioral change and facilitate a more comprehensive assessment of the impact of interventions.

It would also be advisable to consider the cultural and social norms that influence alcohol consumption, especially in populations where BD is an accepted practice.

Another relevant limitation concerns the geographical scope of the included studies. All the interventions analyzed were conducted in high-income Western countries (United States, United Kingdom, and other European nations). Consequently, the evidence base primarily reflects sociocultural and policy contexts characteristic of these regions, which may limit the generalizability of findings to other populations. Future research should therefore aim to test and adapt these intervention strategies in underrepresented regions, including South America, Africa, and Asia, where cultural norms, drinking patterns, and access to preventive resources may differ substantially. Expanding research to these settings would help to ensure that prevention and intervention approaches are globally relevant and culturally sensitive.

Finally, it is essential to promote longitudinal studies with prolonged follow-ups, which allow for the evaluation of the sustainability of effects beyond the short term. The findings of this review suggest that many of the observed effects tend to diminish over time, highlighting the need to include reinforcement components or “booster” interventions to consolidate the achievements made.

Although the interventions reviewed focused on behavioral and motivational components, the 21 RCTs analyzed did not explore in depth the role of underlying neuropsychological factors or the broader social context as moderators or mediators of intervention success. This is a notable omission. Recent evidence highlights that binge drinking, particularly in young people, is associated with specific neuropsychological profiles, such as deficits in inhibitory control, impaired executive functions, and altered reward processing [75,76]. Interventions that do not address these underlying cognitive deficits may be missing a key component for sustaining change. Similarly, although “social norms” were a target of intervention, other deeper social determinants (such as the influence of peer networks, social identity, family dynamics, or socioeconomic status and alcohol accessibility) were not addressed. These factors are critical, as they often shape the ecological context that supports or undermines an individual’s efforts to change [77]. Future interventions could be more effective if they integrated cognitive training components (targeting executive functions) or adopted a broader ecological approach that addresses the individual’s social environment not just their personal motivation.

## Figures and Tables

**Figure 1 life-15-01709-f001:**
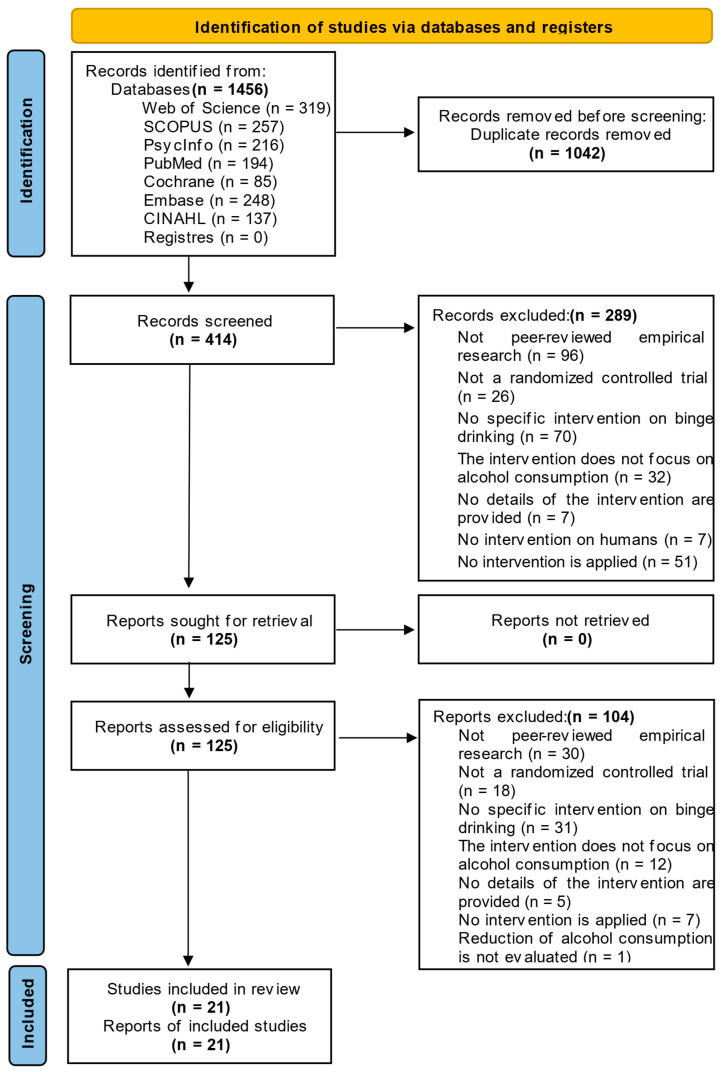
PRISMA 2020 Flow chart summarizing the search process and result [24].

**Table 1 life-15-01709-t001:** Results of the risk of bias assessment (RoB 2).

Study	D1	D2	D3	D4	D5	Overall
Murgraff et al. [27]			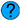	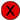		
Wood et al. [28]						
Hedman & Akagi [29]		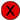	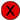			
Conrod et al. [22]	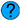					
Daeppen et al. [30]						
Arden & Armitage [31]						
Voogt et al. [32]						
Voogt et al. [33]			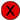			
Witkiewitz et al. [34]				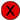		
Voogt et al. [35]						
Williams et al. [36]						
Jander et al. [37]	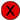		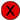			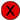
Gilmore & Bountress [38]						
Hanewinkel et al. [26]						
Crombie et al. [15]						
Vargas-Martínez et al. [39]			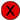			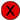
Martínez-Montilla et al. [40]	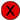					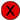
Chavez & Palfai [21]	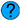					
White & Pohl [41]						
Voss et al. [42]				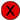		
Carey et al. [11]						

Note: D1: bias due to the randomization process; D2: bias due to deviation from intended intervention; D3: bias due to missing outcome data; D4: bias in measurement of the outcomes; D5: bias in selection of the reported result. 

: low risk; 

: some concerns; 
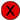
: high risk; 
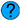
: no information.

## Data Availability

No new data were created or analyzed in this study. Data sharing is not applicable.

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
