# Peer review of "A Systematic Review of Evidence-Based Interventions to Reduce Binge Drinking"

_life, 2025, doi:10.3390/life15111709_

Round 1
Reviewer 1 Report
Comments and Suggestions for Authors
This is a very important, timely, and well done study. BD is a health hazard that had been increasing, particularly in young cohorts and is imperative to provide empirical-based interventions. I am providing several comments and suggestions that, if considered, should imporve significantly the MS.
-The definition of BD in the abstract is vague, does not mention grams of alcohol. This is later fixed in the intro, yet it is important that the abstract has that as well. Abstract: change "Despite numerous interventions" to "Despite numerous interventions to reduce BD".
-The definition of BD in the intro needs a more nuanced explanation. It should indicate how many drinks are the equivalent of the 70/56g indicated. Also, the definition of BD usually includes a specific description of the timeframe of the drinking, oftem within 2 hours. Moreovers, the definition is not static across lifetime, with the boundaries in grams or drink being lower in ages 14-15, usually 3 drinks from girls and 4 for boys. This is important because later the epidemiology discussion in the intro is focused in people 14-15 yrs old.
-It should also be indicated that other terms can be found in the literature for the same behaviors: heavy episodic drinking (HED) or episodic excessive drinking (which are later used in the search strings). These are often defined internationally as ≥60 g of alcohol on one occasion, so more background and citation should be given for the threshold defined by the authors, of 70/56 g. Lastly, it should be noted that BD usually raises blood alcohol concentration (BAC) to about 0.08 g/dL.
-The discussion of the epidemiological data in the intro is solely based in Europe and Asia, there are plenty of other studies in Asia and LatinAmerica. Those should be described to provide a non first-world bias of the intro.
-Please define what a RCT is.
- The filters leading to 21 studies are greatly explained, yet it seems that the restriction of being an RCT (which I assume leaves out those not randomly assigning subjetcs to intervention and control groups) probably moved out a lot of relevant studies. I am not asking to include these in the main analyses, after all the richness of the review is to include only high quality studies, yet there may be valuable studies that were taken out. I suggest including a short section narratively describing "other studies" that were taken out but but are still valuable and can provide material for the aim. I again encourage authors to specifcally value studies outside of Europe-North America, where conducting an RCT study can be nearly impossible due to lack of funding or logistical resources. Yet the knowledge of such studies can still be valuable.
- In 2.3. it is not clear what the "the Risk of Bias 2 (RoB 2) tool" is. Please explain, focusing on the metrics it provides.
-I really like the flow diagram, yet the end of several boxes is cut and text cannot be read, please fix. Perhaps due to do this the n of some records do not add up. For instance, the records excluded after records screened should be 289 but my count gives me 232. Please fix.
-Table 1 is very well done, quite informative, congratulation for the design and the thought behind it.
-The description of the results is very well done and written. I wonder, however, if there was an analysis of whether some of the interventions tackled the problem of potential deceptive responding, a problem that has been tried to adress via control or knowledge-based questions. If possible, comment on this.
-It is not surprising that frequency rather than amount of alcohol was the main variable under analysis. Grams of alcohol consumed is significantly more dificult o measure. Please comment if some studies employed graphical (i.e., posters specyfing the equivalence betweeen commercial drinks and standar units) or specific aids to facilitate the response of the participants.
-Section 3.5.4. Please note that "negative consequences resulting from consumption" is not usually considered a "psychosocial variable", but rather another outcome of drinking or even a proxy of it, that sometimes it is differently modulated by distal and proximal antecedents (i.e., two persons can drink the same but if one executes behavioral protective strategies he/she will show less consequences).
-The dicussion on gender in lines 564-576 seems to miss that participation of women in the 21 studies outweighted males. This should be commented, probably in the broader known scenario that women are much more likely or willing to participate in bomedical research or interventions than men. You could cite examples.
- It is said "Finally, a relevant contribution of the evidence is the identification of side effects in the psychological and social spheres" "...motivation for change and consumption expectations (Gilmore & Bountress, 2016; White & Pohl, 2022); the modification of normative beliefs and the improvement". These are not "side effects" but rather the mechanisms by which the interventions likely operate. And there is plenty of cross-sectional or longitudinal non-experimental research showing that these proximal factors interact with distal, personality like factors, modulating BD. For instance, several studies group hav shown that greater impulsivity is associated with the development of more exagerated social norms about alcohol use which leads to greater use and hence more negative consequences. This section of teh diss needs to dive into this basic science literature that helps understand the efficacy, or lack of there of, the studies reviewed in the paper.
- There is no comment as to wehere the studies come from, albeit country of origin is indeed indicated in the main table. All of them come from USA/UK or Europe. In the future recommendations sections there is no indication of this being a limitation nor the need to expand research to other underserved populations.
Author Response
Hemos incluido respuestas detalladas a cada uno de los comentarios del revisor en el documento adjunto.

Reviewer 2 Report
Comments and Suggestions for Authors
The paper has a great actuality as preventing binge drinking is a real public health challenge in the modern world. Thus, a systematic review of ecidence-based interventions is especially welcome.
Abstract:
Well written, emphasizing the lack of research and what this reviewc can add to the literature. Results are highlighted and evaluated, and future research is recommended.
Introduction:
A clear definition has been provided, however, there are more concepts and definitions which also should be noted here.
I recommend that the authors should also mention difficulties in the development of interventions, especially the cultural background. Alcohol use is so deeply rooted in western cultural norms which makes prevention really difficult.
In addition, the authors should also mention the aims of these interventions as total abstinence is nearly impossible. However, concrete aims should be given in all interventions as they may be different.
The PRISMA model is well presented, although in the literature search strategy the authors use rather broad categories, including rehabilitation and treatment. An important question my arive here: how about the target piopulations? How much are they different?
Table 1 about the risks of bias assessment is really welcome.
Line 341: Theoretical basis, I would suggest to place this part earlier.
Line 477: Discussion: I suggest to divide it into subchapters with relevant heading for greater trasparency.
Strength, limitations and future directions are appropriate.
This study draws the attention to the necessity of evidence based intervention in relation to alcohol use, therefore, I definitely support its publishing after the revision.
Author Response
We have included detailed responses to each of the reviewer’s comments in the attached document.

Reviewer 3 Report
Comments and Suggestions for Authors
The manuscript is methodologically sound and highly relevant; however, several aspects could be improved. The impact of methodological bias and heterogeneity across studies should be analyzed in more depth. The use of RoB 2 is appropriate, but the text only reports proportions (17 studies) without exploring how these biases impacted the findings. The discussion could further explore moderating factors such as age, gender, and cultural context, and more clearly differentiate the magnitude and consistency of intervention effects. The authors focus almost exclusively on psychological interventions but include brief mentions of pharmacological interventions, without developing this contrast. My recommendations:
- Include comparative efficacy tables (e.g., Cohen's d or 95% CI) for each program or theoretical model.
- Add meta-analysis (even exploratory) by type of intervention and age group.
- Deepen critical discussion of publication biases and specific methodological limitations.
- Explore moderating variables (gender, age, culture) with greater emphasis.
- Highlight conceptual gaps, such as the role of unaddressed neuropsychological and social factors.
Author Response

(The authors gave the same response as above.)

Round 2
Reviewer 1 Report
Comments and Suggestions for Authors
The authors have adequately adressed my suggestions.
Reviewer 2 Report
Comments and Suggestions for Authors
I really appreciate that the authors have taken the revision seriously. Overall, all my requests have been met. Therefore, I recommend publishing in this revised form.